# Preparing for the future: The changing demographic composition of hospital patients in Denmark between 2013 and 2050

**Anna Oksuzyan[1]\*, Andreas Höhn[1,2,3], Jacob Krabbe Pedersen[2,4], Roland Rau[1,5], Rune Lindahl-Jacobsen[2], Kaare Christensen[2,4]**

**1** Max Planck Institute for Demographic Research, Rostock, Germany, **2** Department of Epidemiology, Biostatistics, and Biodemography, University of Southern Denmark, Odense, Denmark, **3** Institute of Genetics and Molecular Medicine, University of Edinburgh, Edinburgh, United Kingdom, **4** Danish Aging Research Center, University of Southern Denmark, Odense, Denmark, **5** Department of Demography, University of Rostock, Rostock, Germany

\* Oksuzyan@demogr.mpg.de

## Abstract

### Background

Population aging will pose huge challenges for healthcare systems and will require a promotion of positive attitudes towards older people and the encouragement of careers in geriatrics to attract young professionals into the field and to meet the needs of a rapidly growing number of old-aged patients. We describe the current demographic profile of hospital care use in Denmark and make projections for changes in the patient profile up to 2050.

### Methods

The Danish population in 2013 (N = 5.63 million) was followed up for inpatient and emergency admissions recorded in Danish hospitals in 2013 using population-based registers. We combined age- and sex-specific hospital care use in 2013 with official population estimates to forecast the profile of hospital days up to 2050 with respect to age and sex.

### Results

The total number of hospital days per year is projected to increase by 42% between 2013 and 2050, from 4.66 to 6.72 million days. While small changes are projected for the population aged 0–69, the largest change is projected to occur for the population aged 70+. The 2013 levels were 0.82 and 0.93 million days for men and women aged 70+, respectively. By 2050, these levels are projected to have reached 1.94 and 1.84 million days. While the population aged 70+ accounted for 37.5% of all days in 2013, its contribution is projected to increase to 56.2% by 2050.

### Conclusion

Our study shows one possible scenario for changes in the hospital days due to population aging by 2050: Assuming no changes in hospital care use over the forecast period, the

**Data Availability Statement:** Raw data of the population projections are available online, on the website of Statistics Denmark, http://www.

statistikbanken.dk. The exact file used for analysis can be shared upon request. In addition, the study involves secondary data analysis of existing anonymized, individual-level register data. Data access has been provided via a secure research network. These individual-level-data cannot be shared as any extraction of individual-level data represents a substantial violation of Data Protection Legislation. However, access to these data can be granted through direct application to Statistics Denmark. In order to gain access to these data, researchers need to be affiliated with an authorized Danish research institution and project descriptions must be approved in detail by Statistics Denmark (contact via: https://www.dst.dk/en/kontakt) and the Danish Data Protection Agency (contact via: https://www.datatilsynet.dk/english/contact-us/).

**Funding:** Both grants – 2P01AG031719, US National Institute of Health and "On the edge of societies: Vulnerable populations, emerging challenges for social policies and future demands for social innovation. The experience of the Baltic Sea States (2016-2021)" – were received by Prof. James W. Vaupel, Interdisciplinary Centre on Population Dynamics, University of Southern Denmark (https://portal.findresearcher.sdu.dk/en/persons/jvaupel), who is not a co-author on this paper. The experience of the Baltic Sea States (2016-2021) provided by the Max Planck Society has no number. All funders had no role in the study design, the data collection and analysis, the interpretation of results, the decision to publish and the preparation of the manuscript.

**Competing interests:** The authors have declared that no competing interests exist.

absolute contribution of individuals aged 70+ to the total hospital days will more than double, and the relative contribution of persons aged 70+ will account for nearly 60% of all hospital days by 2050, being largest among men.

## Introduction

Decreasing mortality trends in high- to low- income countries will rapidly increase the population share of older individuals within the next few decades [1]. Currently, in these countries, approximately one person in six is aged 65 or older. By 2050, it is projected that the number of people aged 65+ will almost double within these countries, and that nearly one in three will be 65 or older [2]. As the prevalence of non-communicable diseases increases with age–including cancers [3], circulatory diseases [4], and dementia [5]–population aging presents new challenges for healthcare systems, including hospital settings [6,7]. Preparing the medical workforce for the changing demographic profile of patients has been identified as one of these challenges [8]. Efficient delivery of high-quality care requires an adequate supply of well-trained health workers to meet the needs of the patients. Already now, a number of countries have reported substantial shortfalls in skilled medical workers [9].

A growing body of recent literature has reported that medical students and student nurses tend to have negative attitudes towards older individuals [10–12]. Working with older patients has often been described as a burden and as less satisfying than working with younger patients. Negative attitudes towards the elderly have been shown to be important predictors for the quality of care and treatment [13] and are one main reason why young medical professionals tend to avoid the field of geriatrics [14]. Recent studies suggest that medical students are not aware of who their future patients will be [11]. This is problematic, as aging and retirement of the baby-boom generation will not only accelerate demographic changes, but are likely to leave holes in the medical workforces [15]. Forecasts of future levels of health expenditure are widely available. However, expenditure forecasts tend to be very sensitive to economic shocks and unforeseen changes in costs or new technologies [16,17]. Less research effort has addressed changes in the level and demographic profile of healthcare demands, including hospital care. The profile of future hospital patients depends predominantly on changes in the demographic composition of the population and is therefore easier to forecast because a large proportion of future patients have already been born [18]. Previous forecasts of hospital care demand have often focused on single causes of admission [19], or specific services only, such as long-term or palliative care [20]. Only a small body of literature has studied the effect of population aging on hospital care use patterns at the national level. While a small number of studies exist for Australia and the US [21], very little is known about changes in hospital care demand at country-level, and within the European healthcare context.

In the present study, we describe the current demographic profile of hospital care use in Denmark and estimate the expected number of hospital days by age and sex in Denmark up to year 2050. Our findings provide an empirical basis to suggest that the Danish population will be in need of a healthcare workforce trained particularly in the field of geriatrics.

## Materials and methods

### Linking register data

This study utilizes routinely-collected register data, covering the entire Danish population. We used the unique personal identification number (CPR-Number), which is assigned to all

individuals residing in Denmark, to link records from the National Patient Register (NPR) with data from the Central Population Registry (CPR). The CPR, established in 1968, contains demographic information, such as information on each resident's vital status, sex, and date of birth [22]. The NPR, a population-based register, covers administrative and medical information on all treatments provided in Danish hospitals since 1977 [23]. The NPR data have high levels of completeness and reliability, making them a valuable data source for research. The study involves secondary data analysis, and it was approved by the ethical committee assigned through the Danish National Committee on Biomedical Research and the Danish Data Protection Agency.

### Estimating hospital care use patterns from linked registers

From the CPR, we identified 5,602,628 men and women alive and residing in Denmark on January 1, 2013, who were at risk of being admitted to hospital in 2013. These individuals were followed up for all inpatient and emergency admissions recorded in Danish hospitals between January 1, and December 31, 2013. We summed up the number of hospital days as well as the population at risk by age separately for men and women. We then divided the number of hospital days by the corresponding population at risk to obtain the average, annual number of hospital days per person by sex and age in 2013.

In contrast to inpatient admissions, information on outpatient treatments includes the start date and the end date of the treatment period, but not the number of days an outpatient has received treatment in a hospital. This makes it difficult to assign each patient with a number of hospital days per admission. In addition, obstetrics-related admissions have the potential to introduce a strong sex bias. We therefore excluded both obstetrics-related admissions and outpatient treatments from our main analysis, but included them in sensitivity analyses.

### Population projection by Statistics Denmark

The most recent population projections are provided by Statistics Denmark, the Danish national statistical office [24]. We utilized the estimates covering the period 2018 to 2050. This is a so-called deterministic projection which makes forecasts based on observed trends in fertility, mortality, and migration–the only parameters that can affect the size and structure of a population–up to the middle of the 21st century. The future trajectories of these parameters are assumed to be a reflection of current trends and are based on levels which have been observed in Denmark within the last four years. A systematic and detailed overview of the projection assumptions that underlie these estimations is given in *S1 Table* [25,26].

### Forecasting the demand for hospital care

Lastly, we projected future numbers of hospital days by age and sex using a baseline forecast design [27,28]. We kept the age- and sex-specific hospital care use patterns of 2013 constant throughout the entire study period and combined them with the population projection provided by Statistics Denmark. We multiplied the average, annual number of hospital days per person observed in 2013 by the corresponding population in each year between 2018 and 2050, for single years of age and separately for men and women. As a result, we obtained the total annual number of hospital days by sex and single years of age, which we aggregated into four age groups, separately for men and women: 0–14, 15–49, 50–69 and 70+. Because the number of hospital days is large, the sampling error is very small, and the confidence intervals are hardly distinguishable from the projected line, and thus are not plotted. Preparation of the dataset was done using Stata (Version 14), whereas forecasts and visualizations were produced using R (Version 3.5.1).

## Results

### Hospital care use patterns in 2013

Fig 1 shows the average number of hospital days per person in 2013 by age. The age trajectory of hospital care use is consistently J-shaped among men and women. Within the first year of life, the average number of hospital days per year is 0.9 days for men and 0.7 days for women. From age 1 onwards, the levels decline and reach a minimum of 0.3 days at age 6 among boys and at age 8 among girls. Thereafter, the levels plateau by age 30, at 0.3 to 0.4 days per year. After age 40, the average number of hospital days per year starts to increase steadily with age, to 4.6 days and 3.7 days among men and women aged 90+, respectively. On average, women spend the same number of days in hospital as men between ages 13 and 45. At age 45 and onwards, the annual number of days in hospital is lower among women than among men.

### Changes in the population structure

According to Statistics Denmark, in 2013, the size of the Danish population was about 5.60 million: 0.98 million individuals (17.4%) aged 0–14 years, 2.56 million individuals (45.6%)

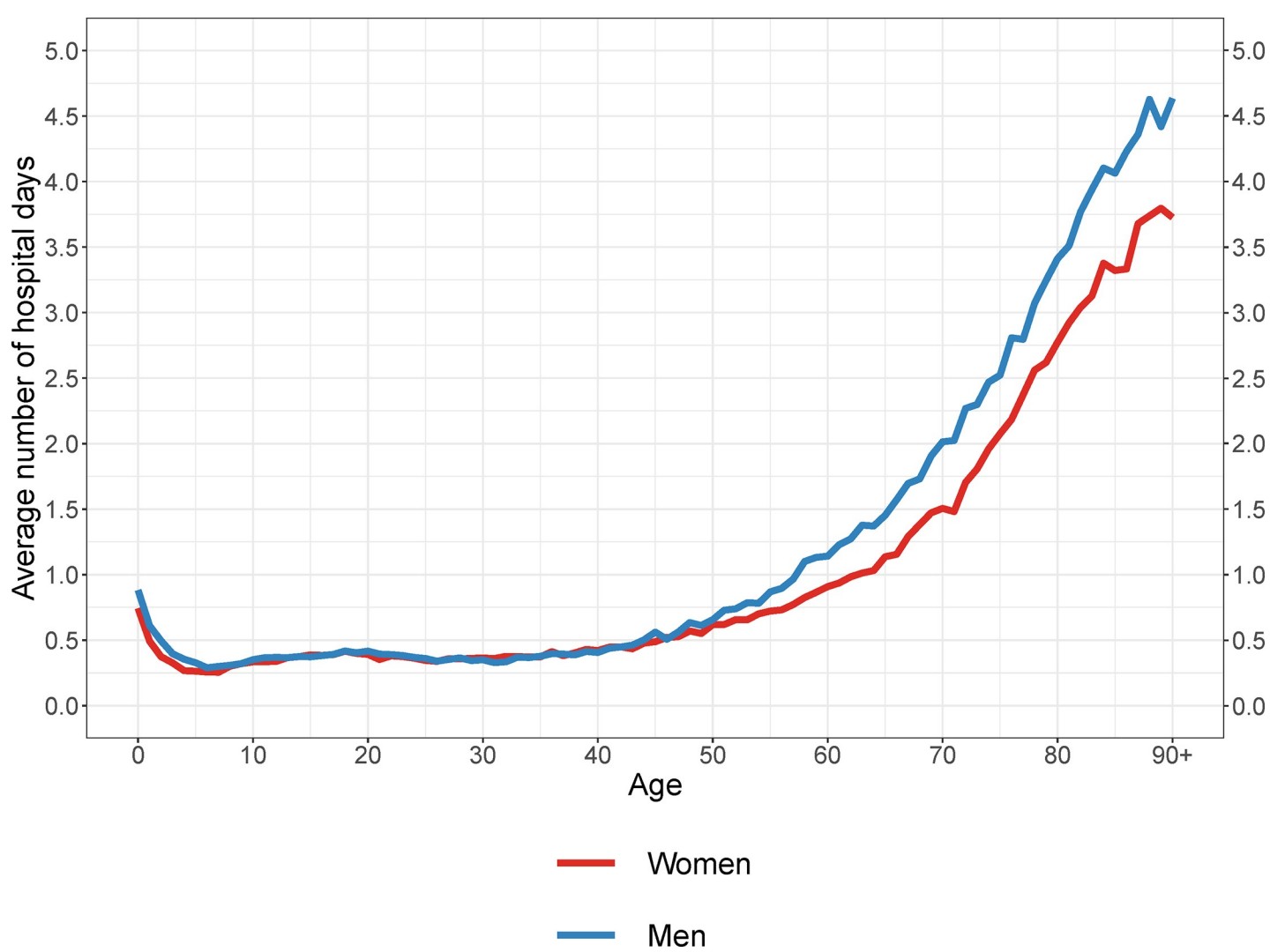

**Fig 1. Average number of days spent in hospital per person in 2013.**

aged 15–49 years, and 1.42 million individuals (25.4%) aged 50–69. In 2013, 0.65 million Danes (11.6%) were aged 70+, of whom 0.28 million were men and 0.37 million were women.

The projections provided by Statistics Denmark up to 2050 are illustrated in Fig 2. It is forecasted that by 2050, the size of the Danish population will increase to 6.43 million individuals. For the population younger than 70, only small changes in size are anticipated: 1.05 million individuals (16.3%) will be aged 0–14, 2.70 million individuals (42.0%) will be aged 15–49, and 1.41 million individuals (21.9%) will be of age 50–69. Major changes are expected to occur in the older population. By 2050, it is projected that 1.27 million Danes (19.8%) will be aged 70 or older, of whom 0.59 million will be men and 0.68 million will be women. That is, the size of the age group 70 years and above will almost double in absolute terms.

### Changes in hospital days

In 2013, we observed 4.66 million hospital days in the Danish population. The population aged 0–14 accounted for 0.37 million days (7.9%), the population aged 15–49–1.07 million days

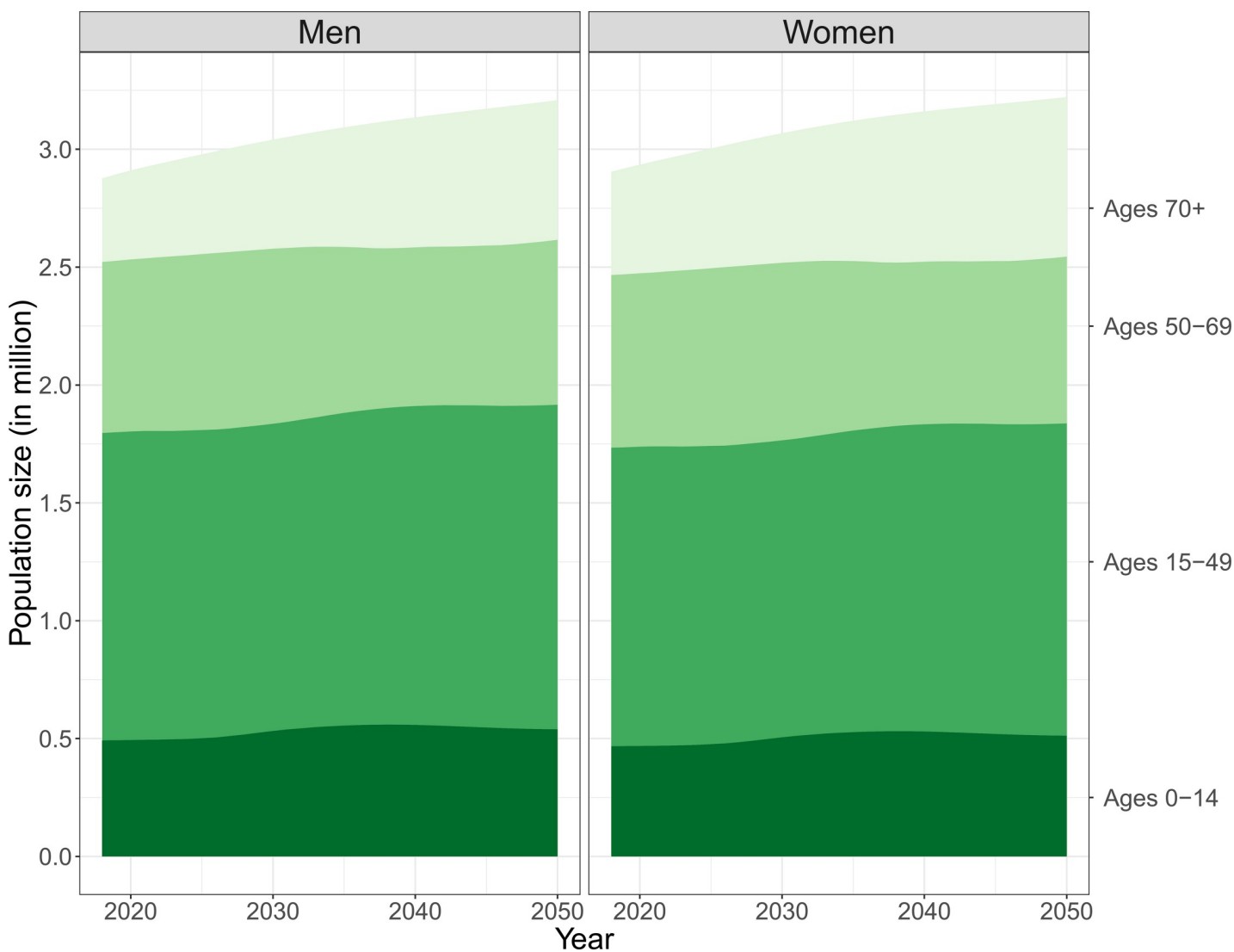

**Fig 2. Population projections by sex between 2018 and 2050 provided by Statistics Denmark.** * Source: https://www.dst.dk/en/Statistik/emner/befolkning-og-valg/befolkning-og-befolkningsfremskrivning.

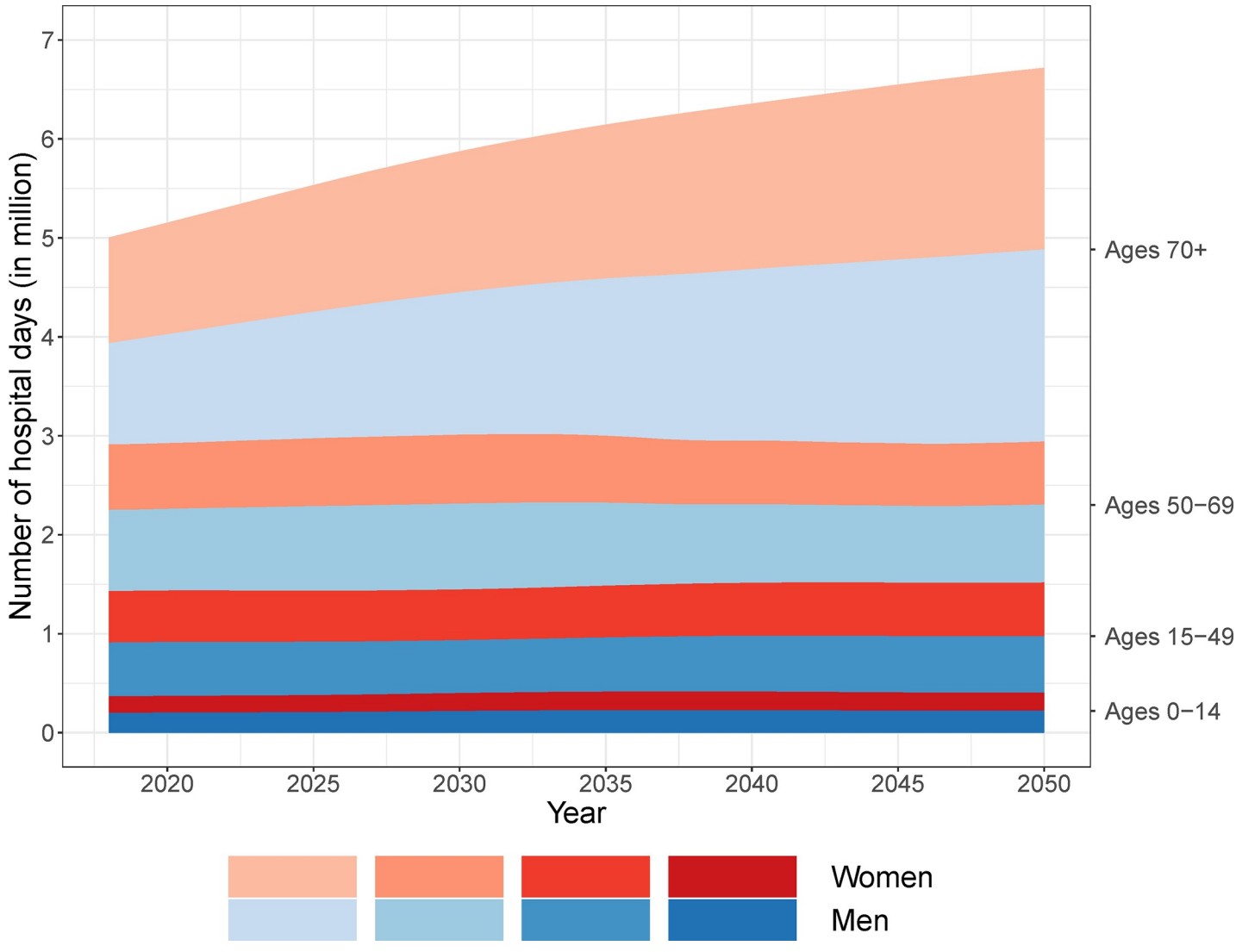

**Fig 3. Projected annual number of hospital days until 2050.**

(22.9%), and the population aged 50–69–1.47 million days (31.6%). In 2013, men and women aged 70+ contributed 0.82 and 0.93 million hospital days, respectively, together accounting for 37.5% of all hospital days.

We combined the age- and sex-specific hospital care use of 2013 with Statistics Denmark's population projections to forecast the number of hospital days up to 2050. Results of the forecast are shown in Fig 3. The total number of hospital days per year increased by 42% within the observed period, reaching an overall level of 6.72 million days in 2050. The number of hospital days in the population younger than 70 is forecasted to remain relatively stable during the projection period: 0.40 million days (6.0%) for the age group 0–14 years, 1.11 million days (16.6%) for persons aged 15–49, and 1.43 million days (21.2%) for the age group 50–69 years. In contrast, the number of days accounted for by the population aged 70+ is projected to increase steadily and to more than double. By 2050, the population aged 70+ is forecasted to account for 3.78 million days, or 56.2% of all hospital days. In the 70+ age group, men are

expected to contribute 1.94 million days whereas women are expected to contribute 1.84 million days. This will correspond to 28.9% and 27.3%, respectively, of all hospital days in 2050.

## Sensitivity analyses

Our main findings do not include outpatient and obstetrics-related admissions. We examined the impact of these admissions on the age- and sex-specific patterns of healthcare use in 2013 and the projection of annual hospital days up to 2050. We considered separately the impact of: (i) admissions due to childbearing and birth control, (ii) outpatient admissions, and (iii) a combination of both. Including these hospital admissions when estimating hospital care use (*further discussed in S1 Text, and* presented in *S1 Fig, S2 Fig, and S2 Table*) did not alter our conclusion that the population aged 70+ remained the most important driver of the increasing number of hospital days over the forecast period. In addition, men aged 70+ were always the fastest growing patient group treated in Danish hospitals in the period up to 2050 in all sets of analyses.

# Discussion

In this study, we show the demographic profile of hospital care use today and in the future. Today, the population aged 70+ already accounts for more than a third of all hospital days. Assuming no changes in hospital care use since 2013 over the forecast period, we showed that the absolute contribution of individuals aged 70+ to the total number of hospital days will more than double and will account for nearly 60% of all hospital days by 2050. By then, men aged 70+ are projected to be the largest patient group treated in Danish hospitals.

## Methodological considerations

We estimated the age- and sex-specific hospital care use for the baseline year 2013 using routinely-collected, individual-level register data. These data cover the total Danish population and minimize biases due to recall or non-response related to under- or overestimation of healthcare use–limitations which often affect studies based on self-reports [29].

Forecasting is, by its nature, uncertain. This applies to two components of our projection: first, the structure of the Danish population and, second, patterns of hospital care use in the future. For the first component we used the most recent population projection of the Danish national statistical office. These projections are deterministic based on a continuation of current trends in fertility, mortality, and migration until the mid-21st century [25,26]. For the parameter with most uncertainty, namely migration [30], Statistics Denmark assumes rapidly declining in-migration in the next few years relative to the 2015–2017 levels, and stabilization of the trend thereafter.

Using a baseline forecast design, we froze the age- and sex-specific levels of hospital care use observed in 2013. Freezing rates assumes that age- and sex-specific hospital care use will remain constant in the future. Therefore, changes in the annual amounts of hospital days during the projection period are driven exclusively by changes in the demographic profile of the population. The S3 and S4 Figs show a summary of admission patterns in Denmark in the years from 2004 up to 2013. The average number of hospital days clearly decline from 2004 to 2013, but less so within the last four years. We performed additional analyses to examine how much the relative increase in the share of hospital days for the age group 70+ at the baseline year influenced the share of hospital days attributed to the 70+ group. Our results suggest that the relative increase in the share of hospital days for the 70+ group is very similar across baseline years varying from 50% (in 2004) to a maximum of 54% (in 2010). As the fluctuations in the age-specific patterns of hospital admissions could have been caused by the individuals'

health behaviours, trends of major chronic conditions, and admission and treatment strategies, long-term predictions of age-specific hospitalization rates are challenging. Keeping baseline levels constant during the projection period is a pragmatic approach in forecasting and has been applied to detailed projections of hospital care use [28] or fertility [31]. Especially when over-arching trends are difficult to predict, freezing rates has been shown to outperform statistically sophisticated techniques [31]. Additionally, the intention of this study is not to accurately forecast hospital care use in future for Denmark, but to provide a glimpse into the relative contribution of population aging to the total hospital care use by 2050, given that hospitalization patterns remain constant over the forecasting period.

## Health and hospital care use in ageing populations

In line with previous findings, our study shows that hospital care use at late and post working ages is especially important for the total national hospital care demand. Previous research has shown that two opposite trends may have an impact on future hospital care use levels at these ages. On the one hand, if incidences of leading causes of death, including stroke and myocardial infarction, continue to decline in low-mortality countries, including Denmark [32], this morbidity postponement may reduce levels of hospital care use among the elderly [33].

On the other hand, it may be that the time spent with major chronic diseases does not shrink as life expectancy increases, and that frailty among the elderly even increases [34]. In addition, medical advances may facilitate safer hospital treatments of older patients, leading to a growing number of individuals at older ages with chronic conditions and multi-morbidity and higher levels of hospital care use in the future [35,36]. However, these new technologies may also enable a shift of treatments from inpatient to outpatient healthcare settings. As all these changes may occur simultaneously, the general direction of population-level trends in hospital care use is difficult to predict. We therefore consider our findings to be neither overly optimistic nor pessimistic, but to reflect a possible scenario.

Apart from long-term trends in admission strategies, hospital care use is also determined by health behaviours. Studies show that smoking, hazardous drinking, obesity, lack of physical activity, and an unhealthy diet are associated with a higher risk of hospitalization at the individual level [37]. General trends in Denmark are encouraging, as smoking rates are decreasing, diet has improved, and physical inactivity over the last two decades has decreased [38,39]. Future levels of hospital care use at the population level will be associated with the age pattern of diseases, trends in health behaviours within populations, as well as with the organizational structure and performance of healthcare systems.

## Implications of a changing patient profile

Our study demonstrates that with the increasing number of individuals reaching older ages, the number of hospital days for individuals aged 70+ will almost double in the period up to 2050.

To meet the complex demands of older patients, a multifaceted approach is needed including changes in training of the medical workforce, organization of the healthcare system, and financial rewards when working with older patients [8,10]. Qualitative studies suggest that medical trainees lack realistic information about the proportion of older patients treated in hospitals and necessary skills to treat these patients, who often have complex health problems intervened with social issues [10,40]. Thus, one key step to move forward to improve geriatric care is through education of healthcare providers [41,42]. Addressing these deficiencies during clinical education may challenge negative stereotypes towards older patients and encourage medical students and student nurses to join the field of geriatrics [12,41]. Furthermore, for

these clinical experiences to effectively enhance students' interest in caring for older patients, clinical placements need to be carried out in the environments that value older patients and are guided by inspirational senior professionals, who appear to be important for junior medical doctors and nurses [10,43].

In Denmark, geriatrics was recognized as a special field in medicine since the 1970s and geriatric patients were treated in hospital long-term care departments or within departments of general internal medicine [42]. In 2008, each of 72 geriatricians served approximately 12,000 persons aged 65 years and over, which is one of the highest ratios across countries included in the European Union Geriatric Medicine Society (EUGMS) [42]. For example, in Belgium, which is the best performing country within the EUGMS, approximately 6,500 adults aged 65+ were served by one geriatrician, whereas the differences in life expectancy in these two countries in 2008 were very small– 0.29 years among men and 0.88 years among women [44]. Currently, geriatric chairs and research centers are located in Aarhus University Hospital, Odense University Hospital, and Aalborg University Hospital with 84 geriatricians and 506 geriatric beds [45]. Given that persons of 70 years and older are the most rapidly growing part of the Danish population (11% in 2013 vs. 19% in 2050), these figures for geriatric services suggest that meeting the medical care needs of these older patients is a complex task, and they highlight an urgent need to attract more junior medical personnel to work in the field of geriatrics. The education of healthcare workers is a long-term process and requires responsible and careful planning. Planning the delivery of healthcare services and preparing for changing care demands require an understanding of who the future patients will be. To meet future hospital care demands, efforts to reduce prejudice towards the elderly among medical students and student nurses must, as soon as possible, be prioritized [12]. Only sufficient numbers and a well-balanced mix of specialists and sub-specialists can ensure that the healthcare challenges of the aging populations will be met in the coming decades [18].

## Conclusion

Already today, older people consume about one third of hospital care (in days), and this share will continue to grow as populations age. If the level of hospital care use remains constant since 2013, the number of days accounted for by the population aged 70+, especially older men, will more than double by 2050. To ensure that hospitals meet the needs of the patient population, a multidimensional approach is necessary. Reducing negative stereotypes about older people and creating incentives to work with older patients, who typically have complex health problems, must be a first step towards ensuring that future doctors and nurses are motivated and qualified to join the field of geriatrics. Taking into account the time frame of training medical staff, these issues should be addressed sooner rather than later.

## Supporting information

**S1 Table. Overview on projection assumptions.**
(DOCX)

**S2 Table. Comparing the impact of different specifications of hospital care use for 2013 (baseline year) and 2050 (last year of the projection period).**
(PDF)

**S1 Text. Further remarks on sensitivity analyses.**
(DOCX)

**S1 Fig. Average number of days spent in hospital per person in 2013 by age and for different admission types and causes.**
(TIF)

**S2 Fig. Projected annual number of hospital days up to the year 2050 for different admission types and causes.**
(TIF)

**S3 Fig. Average number of hospital days per person in the period from 2004 to 2013 by age among men.**
(TIF)

**S4 Fig. Average number of hospital days per person in the period from 2004 to 2013 by age among women.**
(TIF)

## Author Contributions

**Conceptualization:** Anna Oksuzyan, Andreas Höhn, Kaare Christensen.

**Data curation:** Jacob Krabbe Pedersen.

**Formal analysis:** Andreas Höhn, Jacob Krabbe Pedersen.

**Funding acquisition:** Anna Oksuzyan, Kaare Christensen.

**Investigation:** Kaare Christensen.

**Methodology:** Anna Oksuzyan, Andreas Höhn, Roland Rau, Rune Lindahl-Jacobsen, Kaare Christensen.

**Project administration:** Anna Oksuzyan, Roland Rau, Rune Lindahl-Jacobsen, Kaare Christensen.

**Resources:** Anna Oksuzyan, Kaare Christensen.

**Software:** Andreas Höhn.

**Supervision:** Anna Oksuzyan, Roland Rau, Rune Lindahl-Jacobsen, Kaare Christensen.

**Validation:** Andreas Höhn, Kaare Christensen.

**Visualization:** Andreas Höhn, Jacob Krabbe Pedersen.

**Writing – original draft:** Anna Oksuzyan, Andreas Höhn, Kaare Christensen.

**Writing – review & editing:** Anna Oksuzyan, Andreas Höhn, Jacob Krabbe Pedersen, Roland Rau, Rune Lindahl-Jacobsen, Kaare Christensen.

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
