## [Decision Letter · Decision Letter 0]

7 Jan 2020

PONE-D-19-24381

Preparing for the future: The changing demographic composition of hospital patients in Denmark between 2014 and 2050.

PLOS ONE

Dear Höhn,

Thank you for submitting your manuscript to PLOS ONE. After careful consideration, we feel that it has merit but does not fully meet PLOS ONE’s publication criteria as it currently stands. Therefore, we invite you to submit a revised version of the manuscript that addresses the points raised during the review process.

Please see that there are several suggestions that should be faced and responded individually. The rebuttal letter must bring comments on each recommendation/suggestion made by the reviewers.

We would appreciate receiving your revised manuscript by Feb 21 2020 11:59PM. To enhance the reproducibility of your results, we recommend that if applicable you deposit your laboratory protocols in protocols.io, where a protocol can be assigned its own identifier (DOI) such that it can be cited independently in the future. For instructions see: http://journals.plos.org/plosone/s/submission-guidelines#loc-laboratory-protocols

We look forward to receiving your revised manuscript.

Kind regards,

Ricardo Q. Gurgel, PhD

Academic Editor

PLOS ONE

Journal Requirements:

2. In the ethics statement in the manuscript and in the online submission form, please provide additional information about the data used in your study. Specifically, please ensure that you have discussed whether all data were fully anonymized before you accessed them and/or whether the IRB or ethics committee waived the requirement for informed consent. If patients provided informed written consent to have data from their medical records used in research, please include this information.

3. Thank you for stating the following financial disclosure:"NO The funders had no role in the design of the study or in the collection, analysis, and the interpretation of data and results."

Please provide an amended Funding Statement that declares *and fully names all* the funding or sources of support received during this specific study (whether external or internal to your organization) as detailed online in our guide for authors at http://journals.plos.org/plosone/s/submit-now.  

Please include your amended statement within your cover letter; we will change the online submission form on your behalf.

Reviewers' comments:

Reviewer's Responses to Questions

**Comments to the Author**

1. Is the manuscript technically sound, and do the data support the conclusions?

Reviewer #1: Partly

Reviewer #2: Yes

2. Has the statistical analysis been performed appropriately and rigorously? 

Reviewer #1: No

Reviewer #2: Yes

3. Have the authors made all data underlying the findings in their manuscript fully available?

Reviewer #1: Yes

Reviewer #2: Yes

4. Is the manuscript presented in an intelligible fashion and written in standard English?

Reviewer #1: Yes

Reviewer #2: Yes

5. Review Comments to the Author

Reviewer #1: This manuscript is clear and well written. It presents a simple study that applies hospital admission rates of 2014 onto population projection data out to 2050 for Denmark.

The authors justify this simple approach by citing other work that reports more complexity doesn’t necessarily improve forecast models. However, it is noted that the “baseline” forecasting in cited is about staff scheduling in immediate response to patient arrivals with short time frames, therefore this “baseline” methodology may not necessarily translate to this more high level approach.

It would have been useful for readers to have at least a summary of admission patterns in Denmark in the years leading up to 2014. Perhaps there has been striking decreasing or increasing trend in average bed numbers per year for the age and sex stratified population leading up to 2014? Given the extent of Danish hospital records available, why didn’t the authors use data from five to 10 years prior to 2014 to at least justify their assumption that 2014 hospital admissions applied out to 2050 has some validity?

More information is needed in the methodology section. Why are no standard errors presented for forecasted results? How did the authors actually calculate the forecasts? Surely the R program used produced some sort of error values or allowed bootstrapping? Did the authors use the fable package in R? What methodology lies beneath the package they used? Was it a time series model with smoothing and ARIMA modelling or some other approach?

The conclusion of the abstract needs rewriting. It currently only concludes the authors’ opinions and does not conclude anything about the actual results presented.

It could be argued that the population forecast data presented by the authors in the results could mislead readers that it was generated by the authors when it was actually conducted by Statistics Denmark and the authors are simply summarizing their work. Perhaps this section should be in the Introduction instead?

Reviewer #2: Dear authors,

Thank you very much for this article on “Preparing for the future: The changing demographic composition of hospital patients in Denmark between 2014 and 2050”. The results are very important for the future of Health Policies in Denmark, organization of the healthcare system and even for changes in medical training. I have some comments and suggestions to improve the quality of the article. However, I have read this article with great pleasure and appreciation. The requested changes are detailed below:

Abstract: I suggest to replace the terms “medical students” (line 21 and 44) and “student nurses” (line 44) to a more general term like “healthcare workers” or “healthcare providers”.

Objective: The main objective is to describe the current demographic profile of hospital care use in Denmark, project changes up to 2050 and interpret this in light of the attitudes of students in the healthcare workforce. The word “attitudes” is related to a qualitative approach. In addition, this topic was little discussed at the section “discussion”. I suggest to remove the last part of the objective (in abstract and at line 87-88). The suggestion of objective is: describe the current demographic profile of hospital care use in Denmark, project changes up to 2050.

Line 264 – Please write in words (in full) the term “MI”

Discussion of results: Given the results presented, little was discussed about the negative attitudes of students towards older adults. Also, little was discussed about the profile of the future hospitals in Denmark. Some questions are important to discuss in the present study, especially at the topics “health and hospital care use in ageing populations” (line 258) and “implications of a changing patient profile” (line 289):

-The number of health professionals will be sufficient for this new demand? Please provide some data from health care providers in Denmark.

-Is there any health policy to provide curriculum changes or to reduce negative attitudes towards older adults?

-Given the increase in hospital admissions and the number of older people in the future, what can Denmark hospitals and healthcare systems propose? What could be the new profile for these future hospitals? (a) Is there any connection with the health system and primary health care? (b) Hospital plan discharge could be a strategy? (c) New facilities of care for the elderly? (d) Support for caregivers and family members? (e)What could be the economic impact for Denmark Government and for the families?

- There are some repetitive sentences at lines 290-291; 293-294; 297-299. The same ideas were repeated before.

6. PLOS authors have the option to publish the peer review history of their article (what does this mean?). If published, this will include your full peer review and any attached files.

Reviewer #1: Yes: Katrina Spilsbury

Reviewer #2: Yes: Andreivna Kharenine Serbim

---

## [Author Response · Author response to Decision Letter 0]

3 Aug 2020

Response to Reviewers

We thank the two anonymous reviewers for their thoughtful comments about and suggestions for our study. We describe how we responded to each reviewer comment point-by-point below (A stands for Authors, and reviewers’ comments are in bold). 

REVIEWER #1: 

This manuscript is clear and well written. It presents a simple study that applies hospital admission rates of 2014 onto population projection data out to 2050 for Denmark.

The authors justify this simple approach by citing other work that reports more complexity doesn’t necessarily improve forecast models. However, it is noted that the “baseline” forecasting in cited is about staff scheduling in immediate response to patient arrivals with short time frames, therefore this “baseline” methodology may not necessarily translate to this more high level approach.

A: We refer to the study by Vrhovec and Tajnikar 2016, which explored the effects of population ageing on four major part of healthcare services: primary care, secondary care, hospital day-care treatments, and hospitalizations up to years 2025 and 2035. In this paper the authors distinguished hospital day-care treatments – a hospital admission lasting less than a day and does not require overnight stay – and hospitalizations – an admission to hospital for at least one overnight stay. In our study we have considered only hospitalizations. For clarify we have moved the reference by Vrhovec and Tajnikar 2016 to the Forecasting the Demand for Hospital Care subsection where the forecast method is first mentioned. 

It would have been useful for readers to have at least a summary of admission patterns in Denmark in the years leading up to 2014. Perhaps there has been striking decreasing or increasing trend in average bed numbers per year for the age and sex stratified population leading up to 2014? Given the extent of Danish hospital records available, why didn’t the authors use data from five to 10 years prior to 2014 to at least justify their assumption that 2014 hospital admissions applied out to 2050 has some validity?

A: This is a very good suggestion and we have performed the analyses accordingly. The figures R1 for men and R2 for women below shows the average number of hospital days per person in the period from 2004 to 2014. The average number of hospital days clearly decline from 2004 to 2013, but less so within the last four years. This figure made also apparent that the hospital care use is lower in 2014 relative to the last 3-4 years. It is likely to be due to incomplete data registration due to delayed reporting from some regional hospitals. The PIs of this broader project applied to Statistics Denmark to obtain these data towards the end of 2015 (received in 2016), which may be the reason for the data to be incomplete yet. Thus, we decided to take 2013 (which is nearly identical to 2012 and 2011) as the baseline year.

Next, we also fully agree that this is a very simple approach and the forecast results can look very different if hospital care use is not constant over the forecasting period. However, in this study we do not aim to provide accurate forecasts of hospital care use for Denmark by 2050 but we rather would like to provide an estimate of what will be the relative age distribution in the total hospital care use by year 2050 if hospitalization patterns remain constant over the forecast period. This study is motivated by our experiences while teaching medical students, who, at least in their early study years, see themselves treating young patients with single curable disease and, preferably, with a surgical procedure. The reality is very different. The students’ view gradually change when they start working in specific residences after graduating from the university, i.e. at the times when they have already decided about their career path. At the same time, research evidence on this topic to make medical student to reflect on their own views is limited in Denmark. Showing the age distribution of hospital care use by sex today and by 2050, even if this is only one possible scenario, may help challenge students’ stereotypical views on who their patients are likely to be. 

Figure R1. The average number of hospital days among Danish men in the period from 2004 to 2014. 

Following Reviewer’s 2 comment to revise discussion, we also added some figures about the availability of geriatric services in Denmark compared to other European countries. Even if hospital care use continues to decline or will increase by 2025, the number of older people served by one geriatric doctor in Denmark is much more unbalanced than in other European countries: ca. 12,000 per 1 geriatrician in Denmark, 9250 – in Switzerland, 6500 – in Belgium, and 6700 – in Czech Republic (1) although the differences in the life expectancy at birth in Denmark vs. these other countries are not substantial. These figures suggest that there is clearly deficit in the number of geriatricians in Denmark already today, and it is likely to more problematic in 30 years even if hospital healthcare use decline in coming decades. 

Figure R2. The average number of hospital days among Danish women in the period from 2004 to 2014. 

More information is needed in the methodology section. Why are no standard errors presented for forecasted results? How did the authors actually calculate the forecasts? Surely the R program used produced some sort of error values or allowed bootstrapping? Did the authors use the fable package in R? What methodology lies beneath the package they used? Was it a time series model with smoothing and ARIMA modelling or some other approach?

A: We used a simple baseline projection without a model to predict how the number of hospitals days at each age and for each sex might develop in the future. Instead, we “assumed” that the exact same average number of hospital days at each age and for each sex observed in 2013 will apply to each of the years from 2018 to 2050. We are well aware that this is may not be the case. Our aim is not to produce the best future predictions, but rather to illustrate how a specific scenario, the hospitalization pattern of 2013, will develop when entirely driven by the development in the Danish population from 2018 to 2050, as projected by the Statistics Denmark, which is a national statistical office. 

These considerations explain also why we did not provide error bounds to the projections. However, to address the Reviewer’s comment regarding sampling error, below we inserted a version of Figure 3 (in the manuscript) with error bound on the projected average number of hospital days at each calendar year. Since there are no errors for the projected population numbers, which are based on a deterministic model, these numbers are assumed to be fixed. The projected number of hospital days, therefore, is a direct age-standardized rate, and we have provided confidence intervals for these using the approximate bootstrap method described in Swift, M B, 1995, Statist Med, 14:1875-88. Simple confidence intervals for standardized rates based on the approximate bootstrap method). Given the large number of events (hospital days), the sampling error is very small and, in fact, for all projected numbers of hospital days (at each year, for each sex, and each of the four age groups), the bounds of the confidence intervals are nowhere further from the projected number than 0.45%. In the Figure below (revised Figure 3 in the manuscript) the width of the lines correspond to the width of the confidence intervals.

The conclusion of the abstract needs rewriting. It currently only concludes the authors’ opinions and does not conclude anything about the actual results presented.

A: Agree. We have revised the conclusions both in the Abstract and at the end of Discussion sections. 

It could be argued that the population forecast data presented by the authors in the results could mislead readers that it was generated by the authors when it was actually conducted by Statistics Denmark and the authors are simply summarizing their work. Perhaps this section should be in the Introduction instead?

A: We are unsure about moving this part into the Introduction section as it described the data which was used in the present study. We modified the title of this subsection and provided additional link to the Statistics Denmark in the references. 

REVIEWER #2: 

Dear authors,

Thank you very much for this article on “Preparing for the future: The changing demographic composition of hospital patients in Denmark between 2014 and 2050”. The results are very important for the future of Health Policies in Denmark, organization of the healthcare system and even for changes in medical training. I have some comments and suggestions to improve the quality of the article. However, I have read this article with great pleasure and appreciation. 

The requested changes are detailed below:

Abstract: I suggest to replace the terms “medical students” (line 21 and 44) and “student nurses” (line 44) to a more general term like “healthcare workers” or “healthcare providers”.

A: The Conclusion in the abstract is modified to address both Reviewers’ concerns. 

Objective: The main objective is to describe the current demographic profile of hospital care use in Denmark, project changes up to 2050 and interpret this in light of the attitudes of students in the healthcare workforce. The word “attitudes” is related to a qualitative approach. In addition, this topic was little discussed at the section “discussion”. I suggest to remove the last part of the objective (in abstract and at line 87-88). The suggestion of objective is: describe the current demographic profile of hospital care use in Denmark, project changes up to 2050.

A: We modified this parag. completely, see p. 6: 

“In the present study, we described the current demographic profile of hospital care use in Denmark and estimated the expected number of hospital days by age and sex up to the year 2050. Our findings provide a necessary empirical basis to suggest that the Danish population will be in need of the healthcare workforce trained particularly in the field of geriatrics.”

Line 264 – Please write in words (in full) the term “MI”

A: Thank you for spotting this missing explanation. 

Discussion of results: Given the results presented, little was discussed about the negative attitudes of students towards older adults. Also, little was discussed about the profile of the future hospitals in Denmark. Some questions are important to discuss in the present study, especially at the topics “health and hospital care use in ageing populations” (line 258) and “implications of a changing patient profile” (line 289):

-The number of health professionals will be sufficient for this new demand? Please provide some data from health care providers in Denmark.

-Is there any health policy to provide curriculum changes or to reduce negative attitudes towards older adults?

- There are some repetitive sentences at lines 290-291; 293-294; 297-299. The same ideas were repeated before.

A: We revised the Discussion section substantially, particularly subsections “Health and Hospital Care Use in Ageing Populations” and “Implications of a Changing Patient Profile”. 

-Given the increase in hospital admissions and the number of older people in the future, what can Denmark hospitals and healthcare systems propose? What could be the new profile for these future hospitals? (a) Is there any connection with the health system and primary health care? (b) Hospital plan discharge could be a strategy? (c) New facilities of care for the elderly? (d) Support for caregivers and family members? (e)What could be the economic impact for Denmark Government and for the families?

A: Our concern is that answering to some of these broad questions will completely change the intended aim of this study that is to have empirical evidence for changes in the total hospital care use by year 2050 as a consequence of growing number of older people. Although it is only one possible scenario, We believe that these materials can be useful to confront medical trainees with the issues of population aging and make them to reflect on how they view working in the field of geriatrics. 

The Danish government implemented some changes in healthcare and social systems that might have impacted hospitalization rates and length of hospital stay. First, they might have altered the length of hospital stay in recent years in Denmark. As show in the figures R1 and R2 the average number of hospital days per person declined steadily from 2004 to 2010. Second, changes in the admission strategies for speciﬁc causes or treatments are likely to have changed as well. Our earlier study comparing hospital care use in the two cohorts born 10 years apart demonstrated that people born in 1905 had more frequent hospital admissions and surgical procedures, but a shorter length of hospital stay than the 1895 cohort at all ages from 85 to 99 years (2). Finally, in Denmark the total number of hospital beds was reduced from 4.29 per 1000 inhabitants in 2000 to 2.69 in 2013 (https://data.oecd.org/healtheqt/hospital-beds.htm). In fact, the large reduction in hospital beds over the last decades has been done through various financial incentives, especially for those in need of care after discharge. Initially, but with regional variations in the time of implementation, municipalities were forced to pay a day rate for hospital stay for each day a citizen was waiting for discharge after hospital treatment ended. This prompted the municipalities to develop extended home care services, as well as constructing care homes and sheltered housing facilities to accommodate the increasing number of older persons needing personal care. Today, although based on a different calculation method, the principle of a financial incentive applies to all municipalities. 

Denmark has also some regulation to provide support for and encourage adult children to take care of their sick parents or other family members. Such, any person of working age can take a leave to care for a close relative (compassionate leave) at home. Depending on the labor market regulations, these people may be paid full (e.g. often in public service) or partial salary. What proportion of families choose this option is not known and, thus, it is very difficult to predict how these may impact hospital care use in future.

---

## [Decision Letter · Decision Letter 1]

27 Aug 2020

Preparing for the future: The changing demographic composition of hospital patients in Denmark between 2013 and 2050

PONE-D-19-24381R1

Dear Dr. Oksuzyan,

We’re pleased to inform you that your manuscript has been judged scientifically suitable for publication and will be formally accepted for publication once it meets all outstanding technical requirements.

Kind regards,

Ricardo Q. Gurgel, PhD

Academic Editor

PLOS ONE

Additional Editor Comments (optional):

Reviewers' comments:

Reviewer's Responses to Questions

**Comments to the Author**

1. If the authors have adequately addressed your comments raised in a previous round of review and you feel that this manuscript is now acceptable for publication, you may indicate that here to bypass the “Comments to the Author” section, enter your conflict of interest statement in the “Confidential to Editor” section, and submit your "Accept" recommendation.

Reviewer #1: All comments have been addressed

Reviewer #2: All comments have been addressed

2. Is the manuscript technically sound, and do the data support the conclusions?

Reviewer #1: Yes

Reviewer #2: Yes

3. Has the statistical analysis been performed appropriately and rigorously? 

Reviewer #1: Yes

Reviewer #2: Yes

4. Have the authors made all data underlying the findings in their manuscript fully available?

Reviewer #1: No

Reviewer #2: Yes

5. Is the manuscript presented in an intelligible fashion and written in standard English?

Reviewer #1: Yes

Reviewer #2: Yes

6. Review Comments to the Author

Reviewer #1: Due to the sensitive nature of hospital information, the data is not fully available from the authors, but the authors provide information on how to obtain it from the relevant organisations.

Reviewer #2: Dear Authors, thank you for the article's changes. It is very clear and well written. The abstract, the discussion and the conclusions are more connected now.

7. PLOS authors have the option to publish the peer review history of their article (what does this mean?). If published, this will include your full peer review and any attached files.

Reviewer #1: **Yes: **Katrina Spilsbury

Reviewer #2: **Yes: **Andreivna Kharenine Serbim

---

## [Editor Report · Acceptance letter]

1 Sep 2020

PONE-D-19-24381R1 

Preparing for the future: The changing demographic composition of hospital patients in Denmark between 2013 and 2050 

Dear Dr. Oksuzyan:

I'm pleased to inform you that your manuscript has been deemed suitable for publication in PLOS ONE. Congratulations! Your manuscript is now with our production department. 

Kind regards, 

on behalf of

Professor Ricardo Q. Gurgel 

Academic Editor

PLOS ONE